# Feasibility of real-time capture of routine clinical data in the electronic health record: a hospital-based, observational service-evaluation study

Neil Bodagh,[1] R Andrew Archbold,[1] Roshan Weerackody,[1] Meredith K D Hawking,[2] Michael R Barnes,[3] Aaron M Lee,[4] Surjeet Janjuha,[1] Charles Gutteridge,[1] John Robson,[2] Adam Timmis[1,5]

[1]Cardiology, Barts Heart Centre, London, UK
[2]Clinical Effectiveness Group, Queen Mary University of London, London, UK
[3]William Harvey Research Institute Queen Mary University of London, London, UK
[4]Centre for Advanced Cardiovascular Imaging, William Harvey Research Institute, Queen Mary University of London, London, UK
[5]The Farr Institute of Health Informatics Research, University College London, London, UK

**Correspondence to**
Dr Neil Bodagh;
neil.bodagh@nhs.net

## ABSTRACT

**Objectives** The electronic health record (EHR) is underused in the hospital setting. The aim of this service evaluation study was to respond to National Health Service (NHS) Digital's ambition for a paperless NHS by capturing routinely collected cardiac outpatient data in the EHR to populate summary patient reports and provide a resource for audit and research.

**Design** A PowerForm template was developed within the Cerner EHR, for real-time entry of routine clinical data by clinicians attending a cardiac outpatient clinic. Data captured within the PowerForm automatically populated a SmartTemplate to generate a view-only report that was immediately available for the patient and for electronic transmission to the referring general practitioner (GP).

**Results** During the first 8 months, the PowerForm template was used in 61% (360/594) of consecutive outpatient referrals increasing from 42% to 77% during the course of the study. Structured patient reports were available for immediate sharing with the referring GP using Cerner Health Information Exchange technology while electronic transmission was successfully developed in a substudy of 64 cases, with direct delivery by the NHS Data Transfer Service in 29 cases and NHS mail in the remainder. In feedback, the report's immediate availability was considered very or extremely important by >80% of the patients and GPs who were surveyed. Both groups reported preference of the patient report to the conventional typed letter. Deidentified template data for all 360 patients were successfully captured within the Trust system, confirming availability of these routinely collected outpatient data for audit and research.

**Conclusion** Electronic template development tailored to the requirements of a specialist outpatient clinic facilitates capture of routinely collected data within the Cerner EHR. These data can be made available for audit and research. They can also be used to enhance communication by populating structured reports for immediate delivery to patients and GPs.

## BACKGROUND

The electronic health record (EHR) is a longitudinal accumulation of electronic

### Strengths and limitations of this study

► Digital templates for data capture in the electronic health record were tested in real time during routine outpatient consultations confirming their clinical practicality.

► The potential utility of data captured within the templates for audit and research was demonstrated by successful download and aggregated analysis of an anonymised extract.

► Methodology was developed for immediate generation of an outpatient report and its electronic transfer to the referring general practitioner (GP).

► The utility of the outpatient report was examined in a survey of GPs.

► It was a limitation that the GPs we surveyed were restricted to one clinical commissioning group in East London and the response rate of 44% leaves the results prone to response bias.

health information collected during routine healthcare provision.[1] It has the potential to optimise documentation of patient encounters, aid communication between healthcare professionals, improve access to medical information and form a repository of clinical data for use in audit and research.[2] Outpatient consultations are frequent hospital-based clinical interactions and represent an important missed opportunity to contribute clinical data to the EHR. Encounters are usually documented in a dictated clinic letter that is either stored in the paper record or scanned into an electronic repository where it cannot readily be searched or curated. The variable structure and content of the dictated clinic letter undermine its utility for communicating clinical information with previous studies affirming that general practitioners (GPs) prefer structured correspondence about the patients they refer for specialist outpatient

care.[3–6] Communication is further undermined by the inherent inefficiency of the clinic letter which can be highly variable in the time taken to arrive at the address of the referring primary care physician after the index consultation. Appropriate usage of the EHR presents a possible solution to these problems. To our knowledge, the EHR's ability to automatically generate reports of the outcomes of outpatient encounters has not been examined in the UK.

Previous failures to deliver a nationwide EHR within the UK have resulted in severe underusage within the National Health Service (NHS).[7] Underusage has been attributed to a variety of factors that include concerns about disruptions to workflow and difficulties with inputting medical record data.[8] The volume of missing data ensures that few audit and research outputs are based on routinely collected data within the EHR.[9 10] In an effort to rectify the issue, the NHS announced an ambition to become fully paperless by 2020.[11] Crucial to the fulfilment of this ambition will be the development and improvement of EHR systems.

In the USA, there has been a drive to increase usage of the EHR through the 'EHR Meaningful Use Programme'.[12 13] This has led to the development of an electronic after-visit summary which aims to provide patients and their referring clinicians with relevant information and actionable instructions.[14] Studies to date have demonstrated that the after-visit summary is valued by both patients and primary care physicians.[15] The EHR is now increasingly used to generate discharge summaries for patients who have received inpatient care[16] but its use in the outpatient setting for real-time data capture and development of clinic letters has received little attention and we have identified only a single report from a cancer clinic.[17] As far as we are aware, the potential benefits of such a document in a cardiac outpatient setting have not been examined.

The CERNER Millennium (Cerner Millenium, Kansas, USA) EHR system operates in >20 hospital trusts across the UK[18] and is widely installed globally.[19] In the present study, we have used Cerner's Power-Chart application to develop a SNOMED-based electronic PowerForm comprising a user-friendly interface for real-time entry of clinical data during consultation in a general cardiac outpatient clinic. The aims of this study were (1) to test the feasibility of outpatient data capture in digital format for the automatic development of a structured patient report, (2) to examine the effects of PowerForm usage on consultation times, (3) to develop methods for immediate electronic delivery of patient reports to referring primary care physicians, (4) to determine the value of patient reports for improving communication with patients and primary care physicians and (5) to confirm the availability of outpatient data entered into the PowerForm for audit and research.

## METHODS

In presenting this research, we have adhered to Standards for Quality Improvement Reporting Excellence guidelines for reporting new knowledge about how to improve healthcare.[20]

### PowerForm

Technical build experts used the Cerner PowerChart application to develop an electronic template (Power-Form) for clinical data entry according to a strict specification based on the following queries: reason for referral, presenting symptoms, risk factors for cardiovascular disease and hypertension, prior cardiac procedures, examination findings, investigations ordered, diagnosis and problems, cardiac treatment and discharge/follow-up arrangements. The queries were developed by a consultant cardiologist and then modified by consensus of the user group. In order to ensure faithful data entry, standardised responses to the PowerForm queries were listed in drop-down menus or in tabular displays requiring single or multiple tick-box responses. An adaptation of Agile methodology[21] was used in developing the PowerForm in incremental steps which were each tested and modified as necessary before being added to the software bank that contributed to the final product. Ease of use was enhanced by applying conditional logic to guide data entry into those fields relevant for a particular patient. The data captured by the PowerForm populated some of the existing data fields within the EHR such as 'Cardiac Procedures' and 'Diagnosis and Problems' using SNOMED terms throughout while the additional cardiac-specific information populated new fields, further enriching Cerner's digital data repository.

### SmartTemplate

This was developed using the PowerChart application to pull information from the PowerForm into a highly structured 'patient report' that summarised the key clinical findings. The report included the reason for referral, risk factors, vital signs, diagnosis and problems, investigations, treatment, discharge/follow-up arrangements and action points for the referring GP. Again, an adaptation of Agile methodology[21] was used in developing the SmartTemplate which was designed to replace the conventional dictated letter in providing the referring GP and the patient with necessary information about the clinic visit. The view-only report generated within the SmartTemplate was immediately available at the end of the consultation.

### Participants

The outpatient PowerForm was made available to three consultant cardiologists attending a weekly general cardiology clinic in Barts Health NHS Trust. New referrals seen in the clinic from 1 June 2016 to 31 January 2017 were included. Follow-up patients were excluded. Use of the PowerForm was at the discretion of the participating consultants who had the alternative option of making a conventional paper record of the

consultation and dictating a clinic letter to the referring GP. Data were entered into the PowerForm in real time and at the end of the consultation there was the opportunity to present the patient with a printed copy of the patient report.

## Consultation times

These were calculated in a substudy of 44 new referrals seen by one of the participating consultants in seven consecutive clinics. The consultation times for PowerForm consultations were compared with the consultation times taken for conventional paper consultations. Consultation time was defined as the time from arrival of the patient in the consulting room to creation of the patient report, or to completion of the clinic letter dictation at the end of the consultation. Consultation times were manually collected using a stopwatch. The data are reported in minutes as mean consultation time±SD.

## Patient report delivery to GP

The patient report, generated within the SmartTemplate, was designed for electronic delivery to the referring GP. This function was introduced incrementally, starting with Cerner Health Information Exchange (HIE) technology to mirror the patient report across the interface between primary and secondary care. This allowed for inspection of the patient report by the GP in the patient's Egton Medical Information Systems (EMIS) file without true data export. The technology for exporting the patient reports directly into the primary care record was then developed and tested in a separate sample of 64 cardiac outpatients. The technology used the Data Transfer Service (DTS)[22] in 125 local practices that had been appropriately configured. On electronic sign off of the report at the end of the consultation, the DTS delivered it directly into the patient's EMIS file. In those local practices not yet configured for DTS transmission, the reports were delivered by NHS mail.

## GP and patient surveys

In order to obtain feedback about the utility of the patient reports, 210 GPs in the Newham Clinical Commissioning Group were surveyed, using SurveyMonkey software (SurveyMonkey, Palo Alto, California, USA). They were provided with three samples of anonymised patient reports and paired dictated letters and invited to complete the survey by answering questions about their preferences. A paper-based survey was also conducted of 53 patients who were invited to answer questions after their consultation on receipt of their patient report. Participation in both surveys was voluntary and all responses were anonymised. The survey questions are shown in online supplementary tables A1 and A2. An ordinal logistic regression model was used to test whether or not a patient's gender or age group could predict a patient's thoughts on the utility of the patient report compared with the conventional dictated clinic letter. The data were analysed using SPSS for Windows V.17.0 software.

## Extraction of cardiac outpatient data

The clinical dataset recorded within the cardiac outpatient PowerForm was extracted from Trust Data Warehouse SQL server tables using a combination of queries. These data were password protected and stored within the NHS Trust system and were then anonymised. Identifying information was removed, including NHS identifiers, address details, visit dates and the identification of medical staff and centres. Retained demographic data were limited to patient gender, age and ethnicity. These anonymised data were then subjected to the aggregated analysis presented in this work. Data manipulation was performed using the Python software language and visualised using the Plotly framework (Plotly, Montreal, Quebec, Canada).

## Approval

According to institutional policy and the UK Health Departments' Research Ethics Service,[23] this work met criteria for clinical service provision exempt from ethics review.

# RESULTS
## Usage of PowerForms

Among the 695 new-patient referrals who attended the cardiac outpatient clinic during the study period, 594 were seen by participating cardiologists. PowerForms were used in 360 (61%) of these patients, the rate increasing from 42% in June 2016 to 77% in January 2017. The average consultation time, measured in the substudy of 44 patients, was 13.97±3.5 min using the PowerForm (n=22) compared with 14.22±2.95 min using conventional paper documentation (n=22). Individual consultation times are shown in online supplementary table A3.

## Patient reports

Highly structured patient reports (figure 1) were made available on electronic sign off for immediate inspection by the referring GP using HIE technology. Electronic transmission of the patient report, tested in a substudy of 64 consecutive patients, was successful, permitting direct DTS delivery into relevant primary care EMIS files in 29 cases. The remaining patients were referred from practices not currently configured for the DTS mode of transmission and their reports were delivered by NHS mail, either to the practice mailbox (n=22) or to a generic mailbox for postal delivery (n=13).

## GP survey

Of the 210 GPs who were supplied with anonymised paired samples of patient reports and dictated letters, 93 (44%) responded to the survey. We were unable to obtain responses from the remainder of the GPs despite sending further email reminders. Detailed responses are provided in additional material (online supplementary table A1). 74 (80%) GPs found

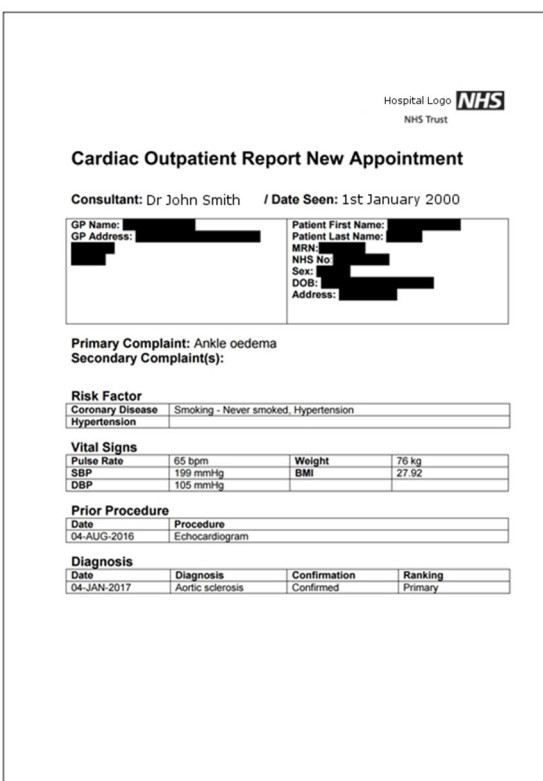

**Figure 1** Sample patient report: sides 1 and 2. BMI, body mass index; CCF, congestive cardiac failure; DBP, diastolic blood pressure; GP, general practitioner; MRN, medical record number; NHS, National Health Service; SBP, systolic blood pressure.

the patient report 'easy to follow', while 69 (74%) considered it provided 'adequate information for their clinical needs'. Electronic transmission into the patient's EMIS file immediately after the consultation was considered important by 75 (81%) GPs, 41 rating it 'extremely important' and 34 'very important'. 70 (75%) GPs found the new patient report more useful than the conventional typed letter, 19 rating it as 'somewhat more useful' and 51 as 'much more useful' (figure 2).

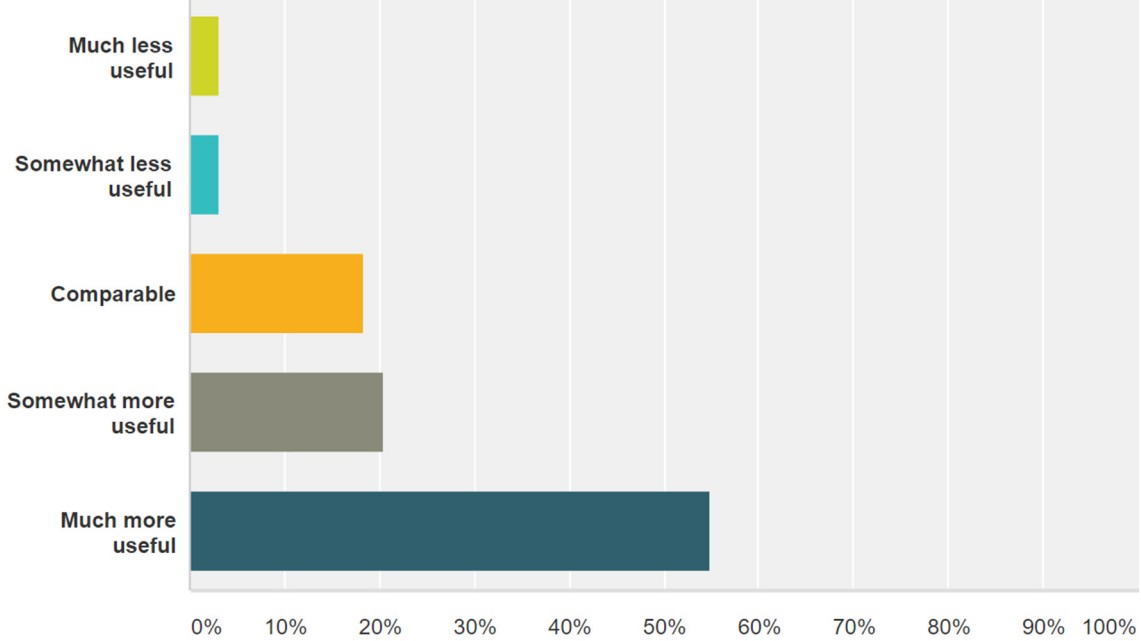

**Figure 2** General practitioner questionnaire. Responses to 'How would you rate the utility of this new Outpatient Report compared with the conventional typed letter posted to your practice?'.

**Table 1** Ordinal logistic regression analysis exploring the relationship between patient demographic variables and their thoughts on the utility of the patient report in comparison with the conventional dictated clinic letter (n=53)

| | | 95% CI | | |
| | OR | Lower bound | Upper bound | P value |
| --- | --- | --- | --- | --- |
| Age | 1.00 | 0.52 | 1.92 | 1.00 |
| Male (ref=female) | 0.85 | 0.27 | 2.73 | 0.79 |

## Patient survey

27 women and 26 men who had been given a copy of their patient reports after outpatient consultation completed the questionnaire. There were no refusals. Detailed responses are provided in additional material (online supplementary table A2). 52 (98%) patients found the report 'easy to follow' and 49 (92%) believed it helpful 'to understand (their) medical condition'. Importantly 49 (92%) patients considered availability of the patient report immediately after the consultation as either 'somewhat more useful' or 'much more useful' than waiting for the postal delivery of a conventional typed letter. Ordinal logistic regression analysis identified no significant relationships between patients' age and sex and their opinions about the utility of the patient report in comparison with the conventional clinic letter (P>0.05) (table 1).

## Data extraction and aggregate analyses

Extraction of data entered into the PowerForm was successful for all 360 patients, confirming availability of these routinely collected outpatient data for audit and research. Sample analyses of reasons for outpatient referral, diagnostic categories and disposal decisions are presented in figure 3.

## DISCUSSION

This study has shown how development of a PowerForm tailored to the requirements of a specialist outpatient service facilitates the capture of routinely collected clinical data within the Cerner EHR. These data can be made available for audit and research. The PowerForm has the potential to enhance communication with primary care physicians and patients by automatically populating structured reports for immediate electronic delivery to the relevant EMIS files and for presentation to patients at the end of the consultation. The clinical utility of these reports was reflected in the surveys we conducted which documented high approval ratings from both primary care physicians and patients. In the present study, we have responded to NHS Digital's ambition for a paperless NHS[11] by designing a PowerForm for routinely collected clinical data in the cardiology outpatient setting that automatically populates a SmartTemplate in producing a novel structured report. PowerForm development within the Cerner EHR provides templates for purposive data entry to meet the requirements of specific clinical tasks.

Our data showed that usage of the PowerForm by participating consultants increased during the course of the study, perhaps reflecting increasing familiarity with its application and an understanding of its added value for communication with primary care physicians and patients. The low rate of PowerForm usage early after its introduction may have been attributable to an initial resistance to a change in work habit, a well-recognised barrier to EHR adoption.[8] Another barrier to EHR adoption is the time burden that is perceived to ensue from its use. However, we found that use of the PowerForm did not prolong consultation times compared with paper-based consultation. For a consultant practised in the use of the PowerForm consultation times were unaffected, and this may help allay concerns that use of the EHR is overly time consuming. Indeed, it is possible to argue that time efficiency is enhanced when the time taken to type, review and mail dictated clinic letters is taken into account.

The immediate availability of the report at the end of the consultation allows a printed copy to be given to the patient as a record of the diagnosis, treatment and further management plan. However, it is the electronic transmission methodology developed as part of this study that represents a true step forward in meeting the ambition for a paperless NHS. In contrast to the traditional clinic letter which may take several days to be typed and mailed to the referring GP, the computer-generated report can now be delivered into the relevant primary care EMIS file before the patient has left the outpatient department.

Our survey showed the value patients placed on this prompt take-home communication that empowered them in helping to understand their medical condition. This was seen as being either very important or extremely important by nearly 90% of GPs who completed the survey. Also favoured was the layout of the report and its structured content, standardising the report of outpatient findings to the referring GP. Indeed, levels of satisfaction with the patient report were high across a variety of domains and it was a major finding in the present study that the majority of patients and GPs found it more useful than the traditional dictated clinic letter for communicating the findings of the outpatient consultation.

Our study confirmed that routinely collected data can be downloaded, anonymised and made available for analysis. The exciting research potential of the EHR has been widely reported[24–28] yet at present the volume of missing data is a major barrier to its use.[9 10] Our study has demonstrated that use of the PowerForm can help to overcome this barrier by capturing routinely entered outpatient data. While the PowerForm's ability to increase EHR-based research output is exciting, its clinical audit function should not be overlooked. The audit exemplars we report—including reasons for outpatient referral, diagnostic categories and disposal decisions—appear mundane but this obscures the fact that information of

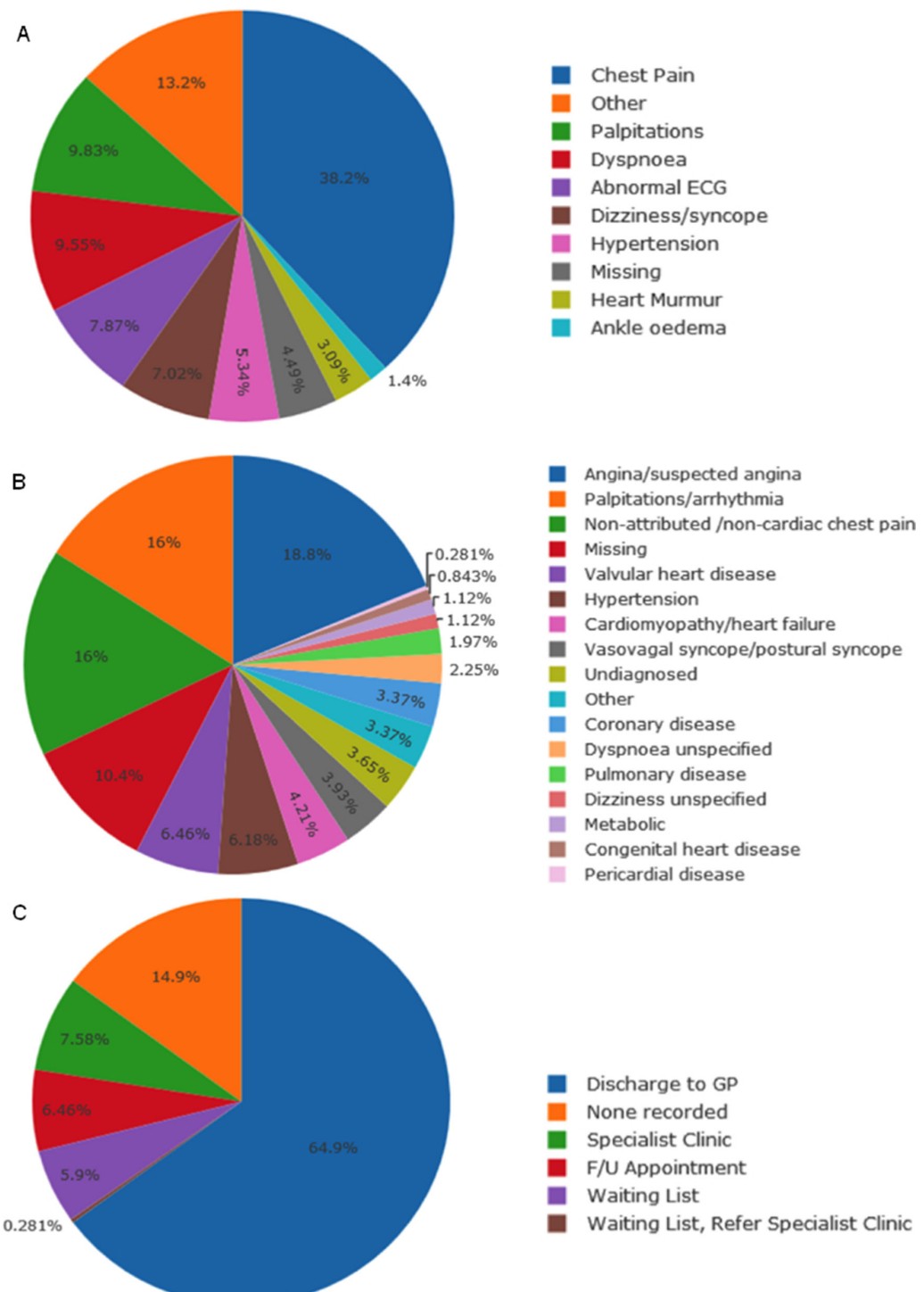

**Figure 3** Aggregate analysis of data extracted from the PowerForm in 360 patients. (A) Indications for outpatient referral. (B) Diagnostic categories. (C) Disposal decisions. F/U, follow up; GP, general practitioner.

this sort is almost never available to clinicians or hospital managers, particularly in settings where clinical documentation is paper based or by free-text entry into computer systems. Understanding these patient profiles seems a basic requirement for effective organisation of outpatient services and provides a compelling rationale for further developing the EHR as a repository for routinely collected clinical data.

The clinical implications of our study are considerable because the technology applied in developing PowerForm and electronic transmission of the patient reports is potentially transportable and available for use in other non-cardiological outpatient settings. Wider development of PowerForms, or their equivalent in other hospital systems, will build up the EHR and provide a substantial data resource for audit and research.

It was a limitation of this study that the GPs we surveyed were restricted to one clinical commissioning group in East London and the response rate of 44% leaves the results prone to response bias. As such, the generalisability of our findings will need testing in larger groups. The same can be said of our convenience sample of 53 patients approached at the end of their outpatient consultation. While there were no refusals, it is important to be aware that these highly positive responses were obtained from a relatively small patient group. Additionally, while the results of this study did not show a significant difference in consultation times between PowerForm and paper-based consultation, comfort using computer systems is variable and the consultation times recorded in this study are not necessarily generalisable to all clinicians. A further limitation was the setting of the study in a general cardiology outpatient clinic and although it is likely that the technology is transportable to other clinical settings, this will need confirmation in future studies. Future studies of live use in other clinical settings will allow further evaluation of the added value offered by the PowerForm. Specifically, studies could be designed to ascertain the usefulness of the patient report as a communication tool between patients and physicians following the index encounter. Studies could also be designed to examine the utility of the patient reports as an educational resource for patients assessing whether patients had referred to them after their outpatient consultation.

## CONCLUSIONS

Using a purpose-built PowerForm, an automated patient report of a cardiac outpatient consultation can be developed providing a feasible alternative to the dictated clinic letter. This report is highly valued by both patients and GPs and is immediately available to both groups. The PowerForm encourages physicians to populate the EHR with coded data during real-time consultation. These data can then be deidentified, downloaded and used for audit and research. Implementation of the PowerForm across other specialties will allow healthcare professionals to capitalise on the benefits offered by the EHR.

**Acknowledgements** AT acknowledges support of Barts Cardiovascular Biomedical Research Unit, funded by the National Institute for Health Research. AT and MRB acknowledge funding from the Medical Research Council. AML acknowledges support from the NIHR Cardiovascular Biomedical Research Unit at Barts Health NHS Trust and from the "SmartHeart" EPSRC programme grant.

**Contributors** NB designed data collection tools, monitored data collection for the whole study, wrote the statistical analysis plan, cleaned and analysed the data, and drafted and revised the paper. RAA and RW designed data collection tools (creation of the PowerForm and usage over the period of the study) and cleaned and analysed the data. MKDH designed data collection tools (surveys of feedback) and cleaned and analysed the data. MRB and JR designed data collection tools and cleaned and analysed the data. AML designed data collection tools (download of data from PowerForms) and cleaned and analysed the data. SJ and CG designed data collection tools (creation of PowerForm and SmartTemplate) and cleaned and analysed the data. AT designed data collection tools, monitored data collection for the whole study, cleaned and analysed the data, and drafted and revised the paper. He is a guarantor. All authors provided intellectual input into the draft of the manuscript, and read, edited and approved the final version.

**Funding** This study was funded by The Guttmann Academic Partnership hosted by UCLPartners. MKDH was in part supported by the National Institute for Health Research Collaboration for Leadership in Applied Health Research and Care North Thames at Bart's Health NHS Trust. NB had financial support from the Isaac Schapera Trust Fund for the submitted work. AML received support from the NIHR Cardiovascular Biomedical Research Unit at Barts Health NHS Trust and from the "SmartHeart" EPSRC programme grant (EP/P001009/1). AT and MRB received funding from the Medical Research Council (MR/K006584/1).

**Competing interests** None declared.

**Patient consent** Detail has been removed from this case description/these case descriptions to ensure anonymity. The editors and reviewers have seen the detailed information available and are satisfied that the information backs up the case the authors are making.

**Ethics approval** According to institutional policy and the UK Health Departments' Research Ethics Service, this work met criteria for clinical service provision exempt from ethics review.

**Provenance and peer review** Not commissioned; externally peer reviewed.

**Data sharing statement** All data generated or analysed during this study are included in this published article (and its supplementary information files).

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
