## [Reviewer comments · BMJ Open]

ARTICLE DETAILS

TITLE (PROVISIONAL)	Feasibility of real-time capture of routine clinical data in the electronic health record: a hospital based, observational service-evaluation study
AUTHORS	Bodagh, Neil; Archbold, R; Weerackody, Roshan; Hawking, Meredith; Barnes, Michael; Lee, Aaron; Janjuha, Surjeet; Gutteridge, Charles; Robson, John; Timmis, Adam

VERSION 1 – REVIEW

REVIEWER	Chris Shea University of North Carolina at Chapel Hill, USA
REVIEW RETURNED	25-Oct-2017

GENERAL COMMENTS	Thank you for the opportunity to review this manuscript, which focuses on the important area of developing more timely and useful information resources for health care providers and patients. Below are areas in which the manuscript could be developed further and/or strengthened. Background 1. Pg 1, Line 14: The authors state: “The EHR, however, is severely under-utilised[3]...” It would be helpful to clarify how it is underutilized. For example, are the authors referring to EHRs not being widely adopted in a specific type of setting? Or are they referring to EHRs being adopted but not used to the fullest extent? Or something else? The citation provided focuses on U.S. hospitals, and the article is from 2009. Are there newer references that indicate the type of underutilization of interest?2. Page 1, Line 27: Please provide a reference to support this statement: “Outpatient consultations are the most frequent hospital-based clinical interactions”3. Have there been evaluations of previous projects using Cerner’s PowerChart Application or of similar efforts to generate reports in other EHRs or in other settings? The citations about the “after-visit” summary are useful, but in general the literature review appears underdeveloped regarding the use of similar methods to develop tools/reports. Without this review, it is difficult for the reader to get a sense of how novel this work is and which aspects of it are most novel.4. The title implies a focus on assessing “benefits for communication” but the four aims are not clearly aligned with that focus. Also notable is that the assessment of consultation times is not an aim, although it could be a primary interest to many readers. The title could include the phrase “feasibility study” as that appears
---

to be an appropriate description.

5. Overall, the Background section could be restructured to frame the study more effectively. After reading this section, the reader should be primed for the four aims. To do so, it would be useful to have a paragraph about the general issue (e.g., opportunities for using EHRs to generate patient reports from outpatient encounters), then a paragraph or two about the specific problems or challenges related to the general issue (e.g., technological feasibility of developing reports, delivering reports, etc.), and finally a paragraph about how this study examines those problems/challenges. Much of this information is in the current version of the Background, but it is not organized in a reader-friendly way.

Methods

1. Page 4, Line 54: How were the queries for the PowerForm decided upon? Was input gathered from stakeholders about the types of information to include in the report?
2. Page 5, Line 25: Please explain in more detail what is meant by “while the additional cardiac-specific information incorporated within the PowerForm further enriched Cerner’s digital data repository”
3. Page 5: A very minor point--I suggest changing the heading “Patients” to “Participants” since the physicians are participants also.
4. Page 7, Line 52: How were consultation times collected?

Results

1. Related to my comment #4 under Background, the results about value for GPs and patients is perhaps the least compelling because of the methods used. For example, it is impossible to know, based on the survey, how the report is useful, why it is more useful than the status-quo “typed letter,” or whether its content is optimal as is or could be improved. Therefore, perhaps the other three aims should be emphasized with the “value” aim should be secondary (rather than primary as implied by the title). This question also relates to comment #3 in the Background section about the underdeveloped literature review. What was the important gap(s) that this project was trying to address? Clearly, the authors cannot go back and redo the survey. However, framing the study more effectively would help set expectations for the reader and also inform how the Results and the Discussion sections should be organized.

Discussion

1. Page 10, Line 46: It’s not clear how the results support the following statement in the discussion: “They can also enhance communication with primary care physicians and patients by automatically populating structured reports for immediate electronic delivery to the relevant EMIS files and for presentation to patients at the end of the consultation.” It’s possible that communication could be enhanced but this wasn’t directly assessed in the study, correct? This point needs to be clarified so that the Discussion does not overstate the results.
2. Page 11, line 21: It seems the word “communication” should be replaced with “report,” “resource” or something similar.
3. The Discussion section (similar to the Background section) could be more reader friendly and ultimately more effective. It should highlight the key findings, put them in context of previous literature, and identify their implications. The second paragraph (Page 11, Line 3-48), in particular, is not well structured. Lines 3-23 are a mix of background and summary of results (e.g., about value for patients). The background information, if general, should be in the first paragraph. If specific to a particular finding, then it should go in the

	paragraph that discusses the implications of that finding. The “electronic transmission method” warrants its own paragraph. 4. Page 11, Line 17-21 The following sentence belongs in the limitations paragraph: “Nevertheless, it should be recognised that comfort using computer systems is variable and the consultation times recorded in this study are not necessarily generalisable to all clinicians.” 5. The Discussion also should clarify implications for future research and practice/management for the key findings. Page 12 Line 35 through Page 13 Line 4, discusses both research and practice implications related to clinical audit. But the two stakeholder groups (researchers and practitioners) are not clearly differentiated. The Discussion should identify specific ways in which the key findings (not just about clinical audit) could lead to better practice/management and inform future research. For example, are there specific research questions that should be addressed in future work on the usefulness the report? 6. Regarding limitations, there should be some recognition of limitations due to the survey items. Is it possible that responses were biased in some way? Also, the survey items are quite general and do not provide much insight into how the report is useful. For example, they do not provide information about whether the report facilitated communication between the patient and physician or whether the patient actually referred to the report either during or after leaving the encounter.
--	---

REVIEWER	theodore pincus Rush University medical Center, Chicago, IL, USA
REVIEW RETURNED	14-Nov-2017

GENERAL COMMENTS	This manuscript presents a well-designed and well-written study indicating that a structured report for outpatient visits to an outpatient cardiology clinical care setting which is available instantly in the electronic records of the primary care physician appears preferred by both patients and doctors to a traditional mailed letter. The results appear predictable, and this reviewer wonders why such this information appears to require an academic report, as an electronic medical record could present such a report as a natural development of the advantages of a computer over paper record. However, since such reports have not been made available using an electronic record, this study presents an advance that might be of interest and value to the medical community. A few specific matters: Page 1 – As noted above, isn’t it almost a tautology that “the report’s immediate availability was considered more useful than waiting for a postal delivery of a conventional typed letter?” Even that were not true, why has email replaced “snail mail” in 98% of applications? Perhaps add “as expected?” Page 3 – Does it really “take many days after the index consultation to arrive...” “is this really the case in the UK? Aren’t the advantages of the structured electronic letter apparent without this comment, which may weaken the argument if exaggerated? Page 5 - Perhaps it would be of interest to know if the some or all the variables, e.g., reason for referral, presenting symptoms, were
---

	available in drop-down menus made up by the reporting physician? That could be helpful to both doctors? Page 7 - Why the alternative of a paper record and dictate a letter instead of the structured letter? If chosen, how often? Why? Page 9 - Why were PowerForms used in 61% of the patients with an increase to 77%? Did some physicians not find the power form of value? Were they older? Why? Page 13 – Same matter - some physicians continue to use the old system – why? One concern is that the technology suggests a “demonstration project” - how generalizable will this report become for others/
--	---

VERSION 1 – AUTHOR RESPONSE

EDITORIAL COMMENTS	RESPONSES – RED FONT REPRESENTS MODIFIED TEXT ENTERED INTO THE MANUSCRIPT
1. Revise title to include research question, study design and setting.	The title has been revised as follows: Feasibility of real-time capture of routine clinical data in the electronic health record: a hospital based, observational service-evaluation study
2. Please revise the Strengths and Limitations section (after the abstract) to focus on the methodological strengths and limitations of your study	They have been revised as follows STRENGTHS AND LIMITATIONS  • Digital templates for data capture in the electronic health record (EHR) were tested in real-time during routine outpatient consultations confirming their clinical practicality • The potential utility of data captured within the templates for audit and research was demonstrated by successful download and aggregated analysis of an anonymized extract • Methodology was developed for generation of an outpatient report that was immediately available for presentation to the patient and electronic transfer to the referring general practitioner • The utility of the outpatient report was examined in a survey of general practitioners. • It was a limitation that the GPs we

	surveyed were restricted to one clinical commissioning group in East London and the response rate of 44% leaves the results prone to response bias.
REVIEWER 1 A. BACKGROUND	
The authors state: “The EHR, however, is severely under-utilised[3]...” It would be helpful to clarify how it is underutilized.	We have amended this section to reflect the fact that we are referring to the EHR being severely under-utilised within the NHS in the UK. We have changed the reference to one of a failed national programme to implement a nationwide EHR in the UK. It now reads: Previous failures to deliver a nationwide EHR within the UK have resulted in severe under-utilisation within the National Health Service (NHS)[7]. Under-utilisation has been attributed to a variety of factors that include concerns about disruptions to workflow and difficulties with inputting medical record data[8]. The volume of missing data ensures that few audit and research outputs are based on routinely collected data within the EHR[9, 10]. In an effort to rectify the issue, the NHS announced an ambition to become fully paperless by 2020[11]. Crucial to the fulfilment of this ambition will be the development and improvement of EHR systems.
Please provide a reference to support this statement: “Outpatient consultations are the most frequent hospital-based clinical interactions”	We have been unable to find a reference to support this statement. We have therefore changed the wording as follows: Outpatient consultations are frequent hospital-based clinical interactions.

Have there been evaluations of previous projects using Cerner's PowerChart Application or of similar efforts to generate reports in other EHRs or in other settings? In general the literature review appears underdeveloped	To our knowledge, similar projects using Cerner PowerChart have not been reported. However, our literature search identified an article describing the generation of a similar report in a cancer clinic. There have also been articles describing generation of standardised discharge summaries within the EHR. The following sentence has been added: The EHR is now increasingly used to generate discharge summaries for patients who have received inpatient care[16] but its use in the outpatient setting for real time data capture and development of clinic letters has received little attention and we have identified only a single report from a cancer clinic[17].
The title implies a focus on assessing "benefits for communication" but the four aims are not clearly aligned with that focus. Also notable is that the assessment of consultation times is not an aim	Title changed: Feasibility of real-time capture of routine clinical data in the electronic health record: a hospital based, observational service-evaluation study Assessment of consultation times now included as a specific aim 2) to examine the effects of PowerForm utilisation on consultation times
The Background section could be restructured to frame the study more effectively.	We have re-structured the introduction as follows:  • Definition of the EHR. Problems with the outpatient encounter at present – no contribution to EHR and communication barrier. EHR has received little usage in the UK for documenting outcomes. • EHR is underutilised in the NHS meaning that routine clinical data are rarely captured and little or no research is generated. The EHR needs to be developed to fulfil NHS Digital's ambition for a paperless service. • Description of the after visit summary and how it would be a potential solution to the problems. Description of use of such a document in a cancer clinic. Not been examined in cardiac outpatient setting. • Aims paragraph.

B. METHODS	
How were the queries for the PowerForm decided upon?	The following sentence has been included to answer this: The queries were developed by a consultant cardiologist and then modified by consensus of the user group
Please explain in more detail what is meant by “while the additional cardiac-specific information incorporated within the PowerForm further enriched Cerner’s digital data repository”	We have clarified this point by modifying the penultimate sentence under the subheading “Powerform” : The data captured by the PowerForm populated some of the existing data fields within the EHR such as “Cardiac Procedures” and “Diagnosis and Problems” using SNOMED terms throughout while the additional cardiac-specific information populated new fields, further enriching Cerner’s digital data repository.
Suggest changing the heading “Patients” to “Participants” since the physicians are participants also.	Done!
How were consultation times collected?	“Consultation time” is defined in the text and we have now added the following: Consultation times were manually collected using a stopwatch.
C. RESULTS	
Results about value for GPs and patients is perhaps the least compelling because of the methods used. The other three aims should be emphasized with the “value” aim secondary (rather than primary as implied by the title	Agreed. The title has been amended accordingly: Feasibility of real-time capture of routine clinical data in the electronic health record: a hospital based, service-evaluation study
What was the important gap(s) in the literature that this project was trying to address?	As previously discussed, the literature around EHR utilisation for capturing routine outpatient data is, we believe, sparse. We have now altered the structure of the Results and the Discussion to reflect our main contributions to the literature, particularly the feasibility of PowerForm as a communication tool for patients and GPs, its impact on consultation times, the availability it gives to data for audit and research and the methodology that was developed for electronic transmission
D. DISCUSSION	

Not clear how the results support “enhanced communication with GPs and patients....” Communication was not directly assessed.	Agreed. The relevant sentence has now been modified as follows: The PowerForm has the potential to enhance communication with primary care physicians and patients by automatically populating structured reports for immediate electronic delivery to the relevant EMIS files and for presentation to patients at the end of the consultation.
Page 11, line 21: The word “communication” should be replaced with “report,” “resource” or something similar.	Done
The Discussion section (similar to the Background section) could be more reader friendly and ultimately more effective. It should highlight the key findings, put them in context of previous literature, and identify their implications. The second paragraph (Page 11, Line 3-48), in particular, is not well structured. Lines 3-23 are a mix of background and summary of results (e.g., about value for patients). The background information, if general, should be in the first paragraph. If specific to a particular finding, then it should go in the paragraph that discusses the implications of that finding. The “electronic transmission method” warrants its own paragraph.	Agree. We have re-ordered the Discussion such that the structure is now:  • Summary • PowerForm utilisation (including consultation times) • Electronic transmission • Survey Results • Data download • Limitations • Further study The aims, methods and results sections are also now in structured in this way (minus summary, limitations and further study).
The following sentence belongs in the limitations paragraph: “Nevertheless, it should be recognised that.....”	Agreed. The sentence has been moved as suggested and modified: Additionally, whilst the results of this study did not show a significant difference in consultation times between PowerForm and paper-based consultation, comfort using computer systems is variable and the consultation times recorded in this study are not necessarily generalisable to all clinicians.

Clarify implications for future research and practice/management for the key findings.	The research potential and audit potential have now been separated. We have also discussed how use of the PowerForm can help alleviate issues regarding missing data within the EHR. The modified text reads as follows: The exciting research potential of the EHR has been widely reported[24-28] yet at present the volume of missing data is a major barrier to its use[9, 10]. Our study has demonstrated that use of the PowerForm can help to overcome this barrier by capturing routinely entered outpatient data. Whilst the PowerForm's ability to increase EHR-based research output is exciting, its clinical audit function should not be overlooked. Specific research questions have been added to the end of the discussion to provide guidance on possible future work that can be done on the usefulness of the report.
There should be some recognition of limitations due to the survey items. Is it possible that responses were biased in some way?	The survey items focus on the utility of the patient report in terms of its content, layout and speed of communication. We agree that there was potential for response bias for the GP questionnaire based on the response rate of 44%. This has been acknowledged in the limitations section. Care was taken to avoid leading question bias in developing the questionnaire.
The survey items are quite general and do not provide insight into how the report is useful. For example, they do not provide information about whether the report facilitated communication between the patient and physician or whether the patient actually referred to the report either during or after leaving the encounter.	Agreed. We have added the following to the Discussion: Specifically, studies could be designed to ascertain the usefulness of the patient report as a communication tool between patients and physicians following the index encounter. Studies could also be designed to examine the utility of the patient reports as an educational resource for patients assessing whether patients had referred to them after their outpatient consultation.
REVIEWER 2	
Isn't it almost a tautology that "the report's immediate availability was considered more useful than waiting for a postal delivery of a conventional	We understand the reviewer's point and have rephrased as follows:

typed letter?" Perhaps add "as expected?"	In feedback, the report's immediate availability was considered very or extremely important by >80% of the patients and GPs who were surveyed. Both groups reported preference of the patient report to the conventional typed letter.
Does it really "take many days after the index consultation to arrive... " Is this really the case in the UK?	Unfortunately this is often the case. However, we do not wish to weaken our argument and have re-worded the sentence as follows: Communication is further undermined by the inherent inefficiency of the clinic letter which can be highly variable in the time taken to arrive at the address of the referring primary care physician after the index consultation.
Perhaps it would be of interest to know if the some or all the variables, e.g., reason for referral, presenting symptoms, were available in drop-down menus made up by the reporting physician?	Agreed – the following has been added In order to ensure faithful data entry, standardised responses to the PowerForm queries were listed in drop down menus or in tabular displays requiring single or multiple tick-box responses.
Why the alternative of a paper record and dictate a letter instead of the structured letter? If chosen, how often? Why?	In providing the alternative of a dictated clinic letter, we were attempting to emulate current practice in the UK where letters are typically dictated with no mandated structure
Page 9 - Why were PowerForms used in 61% of the patients with an increase to 77%? Did some physicians not find the power form of value? Were they older? Why? Page 13 – Same matter - some physicians continue to use the old system – why?	We can't be sure exactly why physicians did not always use the PowerForm, but have included the following statement speculating that The low rate of PowerForm utilisation early after its introduction may have been attributable to an initial resistance to a change in work habit, a well-recognised barrier to EHR adoption[8]. Another barrier to EHR adoption is the time burden that is perceived to ensue from its use. However, we found that use of the PowerForm did not prolong consultation times compared with paper-based consultation. For a consultant practiced in the use of the PowerForm consultation times were unaffected, and this may help allay concerns that use of the EHR is overly time consuming.
One concern is that the technology suggests a "demonstration project" - how generalizable will this report become for others	We recognise the reviewer's concern and are currently engaged in developing use of Powerforms in other clinical settings – with some success. Thus developments are well

	advanced in our "high risk CVD clinic and our heart failure clinic. Meanwhile we have amended the title of the article to include the phrase 'feasibility project' to reflect this.
--	---

VERSION 2 – REVIEW

REVIEWER	Christopher M. Shea University of North Carolina-Chapel Hill
REVIEW RETURNED	29-Jan-2018

GENERAL COMMENTS	Thank you for carefully considering and addressing the previous comments.
---